# Comprehensive Analysis of Physiological, Biochemical and Flavor Characteristics Changes in Crucian Carp (*Carassius auratus*) under Different Concentrations of Eugenol

**DOI:** 10.3390/foods12152820

**Published:** 2023-07-25

**Authors:** Lexia Jiang, Baosheng Huang, Jiaming Tang, Peihong Jiang, Dongjie Chen, Changfeng Zhang

**Affiliations:** 1Shandong Key Laboratory of Storage and Transportation Technology of Agricultural Products, Shandong Institute of Commerce and Technology, Jinan 250103, China; 13977695205@163.com (L.J.); 18396898921@163.com (B.H.); tang19980503@163.com (J.T.); jiangpeihongjph@163.com (P.J.); dongjie613@163.com (D.C.); 2College of Food Science and Technology, Guangdong Ocean University, Zhanjiang 524088, China; 3National Engineering Research Center for Agricultural Products Logistics, Jinan 250103, China; 4Shandong GuoNong Logistics Technology Co., Ltd., Jinan 250103, China

**Keywords:** eugenol, anesthesia, crucian crap, physiological and biochemical indexes, histology, flavor

## Abstract

Eugenol is a widely used fishery anesthetic. This study investigated the effects of various concentrations of eugenol on blood physiological and biochemical indexes, and muscle flavor, in crucian carp (*Carassius auratus*). To determine the appropriate concentration of eugenol anesthetic for use in crucian carp transportation and production operations, we evaluated seven anesthesia groups of 20, 30, 40, 50, 60, 70, and 80 mg/L and one control group (without eugenol) to determine the effects on blood physiological and biochemical indexes, and muscle flavor. The red blood cells and platelets of crucian carp decreased significantly (*p* < 0.05) with eugenol treatment. With increasing eugenol concentration, the white blood cells and hemoglobin did not change significantly, whereas lactate dehydrogenase, alkaline phosphatase, alanine aminotransferase, and aspartate aminotransferase increased significantly (*p* < 0.05). The content of phosphorus, magnesium, and sodium increased after anesthesia, whereas the content of potassium decreased with increasing eugenol concentration. After anesthesia, the content of albumin and total protein in the serum decreased with increasing eugenol concentration (*p* < 0.05); triglyceride first increased and subsequently decreased (*p* < 0.05); blood glucose content first increased and then decreased (*p* < 0.05); and no significant difference was observed in total cholesterol content (*p* > 0.05). No significant difference was observed in muscle glycogen and liver glycogen content after eugenol anesthesia (*p* > 0.05). The eugenol-based anesthesia test did not indicate major liver histomorphological effects, but the very small number of gill sheet edema cases observed requires further study. Analysis of electronic nose data indicated that eugenol treatment affected the flavor of the fish. The anesthesia concentration of 20–80 mg/L had some effect on the physiology and biochemistry of crucian carp, thus providing a reference for the application of eugenol in crucian carp transportation and experimental research.

## 1. Introduction

Crucian carp (*Carassius auratus*) is an important freshwater aquaculture fish in China, because of its high nutritional value, fresh and tender taste, fast growth rate, and strong adaptability [1]. During transportation, crucian carp show an intense stress response. Improper operations can easily cause quality deterioration and death, thus substantially affecting breeding production and transportation. The rational use of fish anesthetics can place fish in a state of anesthesia and sedation, thereby decreasing their consumption of oxygen and energy, and metabolic rate; alleviating their stress response; and facilitating operations and transportation [2]. Simultaneously, the use of anesthetics can decrease activation of the hypothalamic-pituitary–adrenal axis [3,4], diminish the pressure and physical damage caused by operations, and help improve fish welfare.

At present, nearly 30 types of fish anesthetics are available, including ethyl m-aminobenzoate methanesulfonate (MS-222), eugenol, and carbon dioxide (CO_2_) [5]. Before these drugs are used, their efficiency and safety must be assessed to ensure fish safety. MS-222 is the only anesthetic approved by the U.S. Food and Drug Administration (FDA), but its application is not safe, owing to a decrease in pH value and the formation of methanesulfonic acid [6]. The advantage of carbon dioxide over other anesthetics is that it does not require a period of withdrawal, but its anesthetic dose is difficult to control [7]. *Syzygium aromaticum* is a plant of the genus *Syzygium* in the family Myrtaceae, native to the Maluku Islands in Indonesia; its buds are generally known as cloves [8]. Cloves contain 15%–20% volatile oil, and 80% of its components are eugenol. Eugenol is a plant-derived natural compound extracted from cloves; its chemical formula is 2-methoxy-4-allylphenol [9]. Pharmacodynamic studies have indicated that eugenol has sterilization, antioxidative, anesthetic, and analgesic effects [10,11,12]. Local analgesia and anesthesia can be achieved by inhibiting the activity of peripheral nerves. Eugenol, as an anesthetic, inhibits the sensory center of the fish brain and elicits decreases in, or even a loss of, reflexes. Its mechanism of action is divided primarily into three stages: the tactile loss period, the excitation period, and the anesthesia period. The tactile loss period refers to the inhibitory action on the fish brain cortex. The excitation period refers to the action on the basal ganglia and cerebellum, and the anesthesia period refers to the action on the spinal cord [13,14]. The U.S. FDA lists eugenol on the Generally Recognized As Safe (GRAS) list, with an acceptable daily intake (ADI) of 2.5 mg/kg^−1^, as recommended by the World Health Organization [15,16]. Eugenol is usually used as a dental analgesic in China. Eugenol has attracted attention in recent years because of its low cost, high efficiency, safety in live fish transportation, and good solubility in freshwater and seawater. It is widely used in fish production and in scientific experiments, such as broodstock egg collection, live fish transportation, and in vivo surgery [17]. Eugenol is advantageous because it is rapidly excreted from the blood and tissues, without inducing the body to produce mutant substances that have no effect on human health [18]. Eugenol at 40–80 mg/L has anesthetic effects on *Trachidermus fasciatus* [19], and eugenol at 20–70 mg/L has anesthetic effects on *Trachinotus ovatus* [20]. Eugenol has been reported as a fish anesthetic in *Salmon trutta* [21] and *Pterophyllum scalare* [22]. Eugenol effectively decreases stress effects in fish, and increases the transportation density and survival rate; consequently, it is frequently used in the transportation of live fish. Research on eugenol in fish has focused primarily on the anesthetic effects of anesthesia stage, time, and behavior. However, relatively few studies have assessed the specific physiological and biochemical effects in fish. Studies have observed substantial differences in the appropriate anesthetic concentrations of eugenol for different fish species.

## 2. Materials and Methods

### 2.1. Fish and Maintenance

Crucian carp were purchased from the Jinan seafood market (Jinan, Shandong, China). Healthy and active adult fish with an average body length of 16.19 ± 1.58 cm and body weight of 283.33 ± 25.16 g were selected and transported to the National Engineering Research Center for Modern Logistics of Agricultural Products. The fish were temporarily cultured at the National Agricultural Products Modern Logistics Engineering Technology Research Center with an aquatic product temperature control circulating water filtration system. The temporary culture conditions were as follows: they were transported to the Research Center for temporary breeding for 2 weeks, continuously inflated, and fed with commercial compound feed once in the morning and once in the evening every day, and treated with sewage and water exchange. During this period, fish that died or survived in poor condition were removed. Feeding was prohibited 2 days before the start of the experiment. Dissolved oxygen ≥ 7 mg/L, pH 7–8, and water temperature 16.30 ± 2.44 °C.

### 2.2. Anesthetic Agents and Preparation

Eugenol (99% purity) was obtained from Sinopharm Chemical Reagent Co., Ltd., Shanghai, China. The preparation of eugenol anesthetic followed the method of Oliveira et al. with appropriate modifications [23]. The eugenol solution was prepared half an hour before the start of the experiment. The eugenol and anhydrous ethanol (volume ratio 1:9) were mixed to prepare the stock solution. The stock solution was diluted and fully stirred to generate the required concentrations during the experiment.

### 2.3. Sample Collection

All animal experiments were carried out in accordance with the U.K. Animals (Scientific Procedures) Act, 1986, and the National Institutes of Health Guide for the Care and Use of Laboratory Animals (NIH Publications No. 8023, revised 1978). Anesthesia testing was performed according to the method of Favero et al. with appropriate modifications [24]. The mass concentration gradient of eugenol was set to 20, 30, 40, 50, 60, 70, and 80 mg/L, and a control without eugenol was additionally assessed. Five crucian carp were randomly placed in each tank, and no significant difference in fish size was observed among groups. Anesthesia was tested with a single-tailed experimental method; that is, each fish was used in only one experiment. In each experiment, when the fish entered the complete anesthesia period, the judgment standard of the complete anesthesia period was that the experimental fish lost the conditioned reflex ability to the external stimulation, the fish body tilted, did not struggle, the respiratory rate was decreased but regular, and the gill movement frequency was low. Blood samples were collected after crucian carp were maintained under anesthesia for 10 min. The sampling method of blood and tissue was as described by Liu et al. [25]. Five test fish were selected at each blood sampling and quickly fished out of the tank. Using a 5 mL disposable sterile syringe to draw blood from the tail vein of crucian carp, the blood volume is more than 5 mL, and each fish only draws blood once. After the blood samples were placed in a refrigerator at 4 °C for 10 min, some of them were added with EDTA anticoagulant to prevent coagulation, which was used for the determination of blood physiological indexes. The other part was centrifuged at 4 °C, 4000 r/min for 20 min, and the supernatant was stored in the refrigerator at −80 °C for the determination of serum biochemical indexes. The liver and muscle were homogenized with 9 volumes (*w/v*) of sterile saline at 4 °C and centrifuged at 6000× *g* for 20 min at 4 °C. The supernatant was collected to determine glycogen content.

### 2.4. Blood Biochemistry

Red blood cells (RBC), white blood cells (WBC), platelets (PLT), and hemoglobin (HGB) were measured with an XT-1800IV Sysmex se-9500 blood cell analyzer and a kit (Shenzhen Mindray Biomedical Electronics Co., Ltd., Shenzhen, China).

### 2.5. Serum Biochemistry

Serum lactate dehydrogenase (LDH), alkaline phosphatase (ALP), alanine aminotransferase (ALT), aspartate aminotransferase (AST), phosphorus (PO_4_^3−^), potassium (K^+^), magnesium (Mg^2+^), sodium (Na^+^), glucose (GLU), albumin (ALB), total cholesterol, triglyceride (TG), and total protein (TP) were determined with BK-280 automatic biochemical analysis. The kit was purchased from Shandong Boke Biological Industry Co., Ltd., Shandong, Jinan, China.

### 2.6. Glycogen Content Determination

Determination of muscle glycogen content and liver mass fraction was performed with glycogen kits purchased from Nanjing Jiancheng Bioengineering Institute (Nanjing, China).

### 2.7. E-Nose Analysis

According to the research method of Chen et al. [26] with modifications, fish meat was ground with a tissue stirrer, and 2.0 g muscle from the control group and experimental group was weighed, then placed in 10 mL sample bottles, which were sealed. Each sample was analyzed in four replicates. The system contained 18 metal oxide sensors (Table 1). Before the experiment, the measurement parameters of the electronic nose were optimized. According to the response signal of the sensor, the measurement parameters of the electronic nose were as follows: carrier gas flow rate: 150 mL/min, headspace generation temperature: 40 °C, injection volume: 2000 μL, injection speed: 2000 μL/s, headspace generation time: 600 s, data acquisition time: 120 s, and lag time: 400 s. The samples were analyzed with a radar plot and linear discriminant analysis (LDA) to remove the individuals with large differences in the samples.

### 2.8. Histopathology

To further compare the effects of eugenol on the tissues of crucian carp, we examined the histological changes in the gills and liver after 10 min of eugenol anesthesia. We carefully opened the operculum and cut the gills into 5–6 μm thick slices with scissors. The gills and liver were fixed with 10% formaldehyde and washed with 4 °C PBS buffer. Subsequently, the samples were gradually dehydrated with ethanol (70–100%), rendered transparent with xylene, embedded in paraffin, and cut into 5–6 μm thick sections for H&E staining and neutral resin sealing. Optical microscopy was used for staining, observation, and imaging. Images were processed in Image Pro Plus 6.0 software.

### 2.9. Statistical Analysis

All data are expressed as mean ± standard deviation. Statistical analysis of the data was performed in SPSS 22.0 software (Version 22, IBM Corp., Armonk, NY, USA). Based on single-factor analysis of variance, Duncan’s multiple comparison method was used for the analysis. In all cases, the minimum level of significance was set to *p* < 0.05.

## 3. Results

### 3.1. Blood and Serum Indexes in Crucian Carp

#### 3.1.1. Blood Components

After eugenol anesthesia, the RBC concentration in each experimental group was lower than that in the control group (Figure 1A). The RBC content in the control group was (0.25 ± 0.01)10^12^/L. Significant differences were observed between the experimental groups, except the 20 mg/L group, and the control group (*p* < 0.05). The WBC content in the control group after eugenol anesthesia was (824.15 ± 9.03)10^9^/L (Figure 1B). A significant difference was observed between the experimental groups, except the 20 mg/L group, and the control group (*p* < 0.05), whereas no significant difference was observed between the concentration of 30–40 mg/L (*p* > 0.05). The PLT content was lower in each experimental group (*p* < 0.05) than in the control group ((24 ± 1)10^9^/L; Figure 1C). A significant difference was observed between the control group and each experimental group (*p* < 0.05). The hemoglobin content of the control group was (90.3 ± 1.46) g/L. No significant difference was observed between the control group and each experimental group, except the 70 and 80 mg/L groups (*p* > 0.05; Figure 1D).

#### 3.1.2. The Changes in Serum Enzymes

After eugenol anesthesia, the concentration of LDH in each experimental group was higher than that in the control group (Figure 2A). The LDH content in the control group was 531.67 ± 13.67 U/L, and the highest LDH in the 80 mg/L group was 2560 ± 17 U/L. A significant difference was observed between the experimental group and the control group (*p* < 0.05). With increasing eugenol concentration, the content of ALP gradually increased (Figure 2B). The concentration of ALP in the control group was 11.32 ± 3.46 U/L. A significant difference was observed between the control group and the 60, 70, and 80 mg/L test groups (*p* < 0.05) but not between the other experimental groups (*p* > 0.05). The content of ALT in the control group was 8.7 ± 0.61 U/L (Figure 2C). The content of ALT in each experimental group was higher than that in the control group, except for that in the 20 mg/L group, which significantly differed from that in the control group (*p* < 0.05). The content of aspartate aminotransferase (AST) in the control group was 326.83 ± 12.16 U/L, whereas that in each experimental group was significantly higher (*p* < 0.05).

#### 3.1.3. Serum Ion Content

The serum PO_4_^3−^ concentration in crucian carp was 4.07 ± 0.16 mmol/L in the control group and was higher in each experimental group after anesthesia (Figure 3A), except for the 50, 60, and 70 mg/L groups, which significantly differed from the control group (*p* < 0.05). The serum K^+^ concentration in crucian carp in the control group was 5.17 ± 0.17 mmol/L. The serum K^+^ concentration in each experimental group after anesthesia was lower than that in the control group (Figure 3B). A significant difference in serum K^+^ concentration was observed between the experimental group and the control group (*p* < 0.05). The concentration of Mg^2+^ in the serum in the control group was 1.60 ± 0.13 mmol/L, whereas that in each experimental group after anesthesia was higher (Figure 3C). A significant difference in serum Mg^2+^ concentration was observed between the experimental group and the control group (*p* < 0.05), and the concentration of Mg^2+^ in the 80 mg/L concentration group was highest, at 2.67 ± 0.08 mmol/L. The serum Na^+^ concentration in the control group was 82.77 ± 4.39 mmol/L, and was higher in each experimental group after anesthesia (Figure 3D). A significant difference in serum Na^+^ concentration was observed between the experimental group and the control group (*p*< 0.05), but no significant difference was observed between the 40–80 mg/L concentration groups (*p* > 0.05).

#### 3.1.4. The Serum Concentrations of Organic Components

The concentration of ALB in the control group was 29.58 ± 3.25 g/L, whereas that in each experimental group was lower (Figure 4A). A significant difference in ALB concentration was observed between the control group and each experimental group (*p* < 0.05). The TP concentration in the control group was 52.87 ± 0.87 g/L, and a significant difference was observed between the control group and each experimental group (*p* < 0.05). The concentration of TG in the control group was 3.15 ± 0.3 mmol/L, and that in each experimental group was higher than that in each experimental group (*p* < 0.05; Figure 4B). No significant difference was observed between the concentrations of 70–80 mg/L (*p* > 0.05). The serum GLU concentration in crucian carp in the control group was 5.9 ± 0.11 mmol/L, and was higher in each experimental group (*p* < 0.05; Figure 4D).

#### 3.1.5. The Glycogen Index

The content of liver glycogen in each experimental group was higher than that in the control group (*p* < 0.05). The muscle glycogen content significantly increased in crucian carp after being anesthetized with different concentrations of eugenol (*p* < 0.05; Figure 5B). The muscle glycogen content of crucian carp in the control group was 3.12 ± 0.42 mg/g, which was similar to the change trend of liver glycogen. A significant difference in muscle glycogen content was observed between the control group and each experimental group (*p* < 0.05), whereas no significant difference was observed among experimental groups except the 20 mg/L group (*p* > 0.05).

### 3.2. Pathology

#### 3.2.1. Gills

The gill tissue sections of the control group and carp anesthetized with different concentrations of eugenol are shown in Figure 6. In contrast with the control group, the 20, 60, 70, and 80 mg/L eugenol-treated groups showed edema and elevation of the gill lamellae, which were not neatly arranged (neither perpendicular to the gill filaments nor parallel to one another), and the bases of the gill lamellae were elevated and not smooth. The mitochondria-rich cells were smoothly distributed at the base of the gill lamellae, with no clear elevation or detachment; the flat epithelial cells were neatly arranged on the gill lamellae; and the overall structure showed normal physiology.

#### 3.2.2. Liver

The liver tissue sections in the control group and groups anesthetized with different concentrations of eugenol are shown in Figure 7. The liver samples of all anesthetized groups were compared with those in the control group, and no significant changes in normal liver histology were observed. Most nuclei appeared in the centers of the liver cells, and prominent heterochromatin and nucleoli were observed. In all experimental groups, no bleeding symptoms, necrotic areas, inflammation, edema, or granuloma were observed. The islets contained acinar cells and many secretory granules.

#### 3.2.3. Muscle

The muscle tissue sections of the control group and the group anesthetised with different concentrations of eugenol are shown in Figure 8. Muscle is composed of many muscle fibers. In the control group, the muscle fibers were closely arranged. The muscle fiber sizes among the concentration groups differed, the muscle fiber arrangement was disordered, and the muscle fiber diameter was significantly smaller than that in the control group. In the 20–80 mg/L concentration group, the muscle fiber gap was further widened, the muscle fiber was disordered, and a clear fracture phenomenon occurred.

### 3.3. Response of an Electronic Nose to Volatile Flavor Compounds in Crucian Carp Anesthetized with Different Concentrations of Eugenol

#### 3.3.1. Radar Fingerprinting of the Effects of Different Concentrations of Eugenol on Muscle Flavor in Crucian Carp

Radar fingerprints of the effect of muscle flavour in the control group and the group anaesthetised with different concentrations of eugenol are shown in Figure 9. The aroma of muscle tissue from crucian carp anesthetized with different concentrations of eugenol was analyzed with an electronic nose. A radar map was constructed to reflect the effects of different concentrations of eugenol on volatile components in muscle. The 18 sensors of the FOX4000 electronic nose differ in their sensitivity to gas. When the sensor is in contact with gas, the ratio of relative conductivity G/G_0_ is proportional to the gas concentration. When G/G_0_ > 1, the gas concentration is high. When G/G_0_ ≤ 1, there is no response gas, or the gas concentration is below the detection limit [27]. As shown in Figure 6, the 18 sensors of the electronic nose responded to different concentrations of eugenol among the volatile flavor substances of crucian carp muscle, and the response intensity differed. The G/G_0_ of T30/1, P10/1, PA/2, P30/1, and P40/2 sensors was higher than that of the other sensors. The concentration of eugenol in the control group was higher than that in all experimental groups except the 80 mg/L group. The effects of eugenol on the muscle in crucian carp in the other experimental groups were unclear, and the difference in the response signal intensity was small. The principal component analysis diagram can be visually distinguished.

#### 3.3.2. LDA Analysis Based on the Electronic Nose Method

The LDA analysis of muscle based on the electronic nose method in the control group and the anesthesia group using different concentrations of eugenol in Figure 10. LDA is a statistical method that uses samples of known categories to establish a discriminant model and discriminate among unknown categories of samples [28]. The electronic nose data after LDA dimension reduction analysis are shown in Figure 7. The contribution rates of LDA1 and LDA2 were 74.2% and 14.25%, respectively, and the cumulative contribution rate was 88.45%. These two principal components essentially reflected all information characteristic of crucian carp. LDA distinguished each crucian carp muscle test group exposed to different concentrations of eugenol and the control group, and showed no overlapping area.

## 4. Discussion and Conclusions

Blood physiological indexes are closely associated with metabolism, nutritional status, and diseases in fish [29]. The main role of WBC is to protect the body and resist invasion pathogens [30]. RBCs are oxygen-transporting cells in the blood, which transport fresh oxygen to the body tissue through the blood in the circulatory system in fish, and simultaneously squeeze the capillaries of the body and transport carbon dioxide from tissue to the lungs, thus allowing the fish to breathe [31]. PLT are small pieces of cytoplasm that are lysed from the cytoplasm of mature megakaryocytes in the bone marrow. These blood cells maintain hemostasis in the body and have active immunomodulatory effects. HGB is a metal protein in red blood cells, which usually transports oxygen and carbon dioxide in fish blood [32]. Liang et al. [33] analyzed the anesthetic blood indexes of clove oil on tilapia (*Oreochromis mossambicus*) and found higher content of WBC, RBC, HGB, and PLT in the blood after anesthesia than in the control group. In our study, the content of RBC, WBC, PLT, and HGB in crucian carp after anesthesia with different concentrations of eugenol was lower than that in the control group. We speculated that eugenol anesthesia might have influenced blood function in fish, in contrast to the results in tilapia research. It is speculated that the results may be caused by temperature or species differences.

Changes in serum enzymes can reflect the metabolism and material transformation of the body, and the different states of tissue structure and function, and serve as important indicators of the integrity of the cell membrane [34]. LDH is present primarily in myocardial cells, and its main function is to catalyze the oxidation of lactic acid to pyruvate in cells. An increase in LDH activity reflects the degree of damage to the kidney of the myocardial cell nucleus [35]. ALP is directly involved in the transfer and metabolism of phosphate groups in organisms, and thus plays a role in metabolism and immune protection in organisms [36]. ALT and AST are important aminotransferases widely present in animal cell mitochondria, and their content is high in hepatocytes [37]. Under normal circumstances, only small amounts of transaminases in hepatocytes are released into the blood, and consequently, the activity of transaminases in the serum is low. When the content of ALT and AST in the serum increases, hepatocytes are considered to be damaged, and ALT is an important indicator of liver damage. In general, the concentrations of LDH, ALP, ALT, and AST in the serum of crucian carp increased after eugenol anesthesia, possibly because of the synthesis and release of enzymes in the body of crucian carp in response to the stress of eugenol anesthesia, thus increasing the activity of these enzymes. This finding is consistent with research results on *Oncorhynchus mykiss* anesthetized with myrcene and eugenol [38].

Inorganic ions such as PO_4_^3−^, Mg^2+^, K^+^, and Na^+^ in the serum play important roles in maintaining plasma osmotic pressure, acid–base balance, and the stability of the internal environment in fish [39]. The kidneys are associated with the metabolism of magnesium and phosphorus. When the kidneys are damaged, the level of magnesium and phosphorus metabolism in fish becomes abnormal [40]. Feng et al. [41] have found that the serum phosphorus and magnesium levels of juvenile Siberian sturgeon (*Acipenser baerii*) increase after eugenol anesthesia, similar to the results of this study. When the concentration of eugenol was 80 mg/L, the content of magnesium reached a maximum. We speculated that with increasing eugenol concentration, the damage to the fish kidneys caused by the anesthetic was aggravated, thus resulting in renal dysfunction, abnormal magnesium metabolism, and serum magnesium retention and increased level. In general, the permeability of fish gill epithelium was greatest to hydrogen ions, followed by water and Na^+^. We inferred that eugenol anesthesia in crucian carp increased the permeability of the gill epithelium. Excessive Na^+^ entered the blood, and its concentration significantly increased, whereas K^+^ was inhibited, and its concentration significantly decreased. This finding is inconsistent with that reported by Hu et al. [42], who have found that the serum K^+^ level in silver carp (*Hypophthalmichthys molitrix*) increases by four times after CO_2_ anesthesia. Therefore, differences in the mechanisms of various anesthesia methods may exist among fish species, and the physiological functions of serum ions with different anesthesia methods must be further explored.

Blood GLU provides energy to support various life activities in fish and is susceptible to fluctuations in environmental factors [43]. The normal blood GLU content in fish is 2.78–12.72 mmol/L [44]. The blood GLU concentration in crucian carp in the control group was lower than that in each group after anesthesia. This finding might have been due to a decrease in blood GLU metabolism after anesthesia, thereby increasing the blood GLU concentration. The physiological functions of ALB in the blood include maintaining colloid osmotic pressure, and transporting serum calcium ions, unconjugated bilirubin, free fatty acids, and hormones [45]. After anesthesia with eugenol, the ALB value in crucian carp decreased, possibly because ALB, as a carrier of nutrients, increased consumption when dealing with the anesthesia slope, thus providing energy for the body, repairing damaged tissues, and participating in maintaining plasma colloid osmotic pressure balance. This possibility is consistent with the results of Zhu et al. [46], who observed the decreased effects of electrical anesthesia on ALB in the blood of juvenile *Coreius guichenoti*. Lipids in the blood serve as raw materials for the metabolism of liver cells, including total cholesterol, triglycerides, high-density lipoprotein, and low-density lipoprotein [47]. Total cholesterol is not only an important component of biofilm but also a prerequisite for the synthesis of bile acids, steroid hormones, vitamin D_3_, and other physiologically active substances [48]. Triglyceride, total cholesterol, and total protein levels are affected by protein catabolism and hepatic glycogen decomposition. In this study, the serum triglyceride content in crucian carp in the experimental groups was lower than that in the control group, whereas the total cholesterol content in the experimental groups did not significantly differ from that in the control group, in agreement with the results of eugenol anesthesia in short-tailed bass (*Piaractus brachypomus*) [19]. The reason for the decrease may be that under anesthesia, blood lipids are involved in metabolism and are transported and decomposed. Serum GLU in fish is generally provided by liver glycogen and muscle glycogen [49]. Muscle glycogen, an important energy source for maintaining the body’s metabolism, exists primarily in muscles and is an important indicator of the degree of environmental stress. Changes in muscle glycogen clearly reflect the body’s adaptation to the environment [50]. The increase in muscle glycogen and liver glycogen concentration in the anesthesia group was much higher than that in the control group, possibly because of the decrease in metabolic rate and the accumulation of muscle and liver glycogen after anesthesia. No significant difference was observed in muscle glycogen and liver glycogen in each eugenol concentration group, thus indicating that different concentrations of eugenol had little effect on the muscle and liver in crucian carp.

The gills are involved in gaseous exchange, acid–base balance, ion balance, nitrogenous waste excretion, osmotic adjustment, and other multi-functional tissues [51,52]. The gills are also a biological indicator used to measure water quality [53,54] and an important means through which fish interact with the external water environment. In addition, fish gills are an important organ for the absorption of anesthetics. The gills are the main respiratory organ in fish, and are responsible for excreting metabolic wastes, such as ammonia nitrogen, and regulating osmotic pressure [55]. The fish gills are composed primarily of gill filaments and gill lamellae. The gill filaments extend vertically and have semicircular flat cystic gill lamellae, which are arranged in parallel. The main part of the gill filament contains gill cartilage, central venous sinus, and gill filament epithelial cells (composed primarily of chloride cells, mucous cells, flat cells, and undifferentiated cells). The gill lamellae are composed primarily of columnar cells, flat cells, and blood vessels (with blood cells). MS-222 has been shown to cause swelling of gill tissue [56], thereby hindering oxygen uptake by gill lamellae. Oliveira et al. [57] have found that eugenol can cause mild lamellar epithelial hyperplasia in the gill tissue in freshwater fish. Similar phenomena were also observed in this study. The liver is an important gland, and digestive and metabolic organ, in fish. It is one of the most important organs responsible for maintaining physiological functions, including bile secretion, metabolism, detoxification, and defense [58]. In this study, the different eugenol anesthesia concentrations in crucian carp did not cause liver tissue damage, in agreement with findings by Velisek et al. [59], indicating that eugenol anesthesia in carp did not cause liver or kidney tissue damage.

The electronic nose is an array based on gas sensors, which simulate the human olfactory system to capture the characteristic aroma of a sample. It has the advantages of low cost, easy operation, and high accuracy [60]. The electronic nose technology based on a bionic olfactory mechanism has developed rapidly in recent years. Li et al. [61] have quantitatively analyzed the freshness of fish meal with electronic nose technology combined with chemometrics. Zhang et al. [62] have used an electronic nose and other technologies to study the volatile flavor components and their production mechanisms in *Trachinotus blochii* fillets treated with different drying methods. Li et al. [63] used an electronic nose to detect the biochemical indexes of *Trachurus japonicus* during frozen storage, to assess its freshness. In this experiment, we observed no overlap between the control group and the groups subjected to different concentrations of eugenol anesthesia; however, overlap existed between the experimental groups, possibly because of the minor effects of eugenol on the flavor substances in crucian carp muscle. The effects of eugenol treatment on the flavor of crucian carp meat may be short-term, and the metabolism of eugenol in fish will continue to be studied in the future.

## 5. Conclusions

According to the changes in blood physiological and biochemical indexes in crucian carp after eugenol anesthesia, we determined that 20–80 mg/L concentrations of eugenol had relatively small effects on the physiological and biochemical indexes, but relatively greater effects on muscle, and did not result in major liver histomorphological effects. However, the edema observed in a very small number of gill lamellae requires further study to determine any potential histological effects of this anesthesia method. In fish production and scientific research, eugenol can be used reasonably in this concentration range, as appropriate. Eugenol affected the original muscle flavor of crucian carp. In the future development of fishery anesthetics, more products with less influence on flavor should be considered. This study provided a reference for the application of eugenol in crucian carp transportation and experimental research.

## Figures and Tables

**Figure 1 foods-12-02820-f001:**
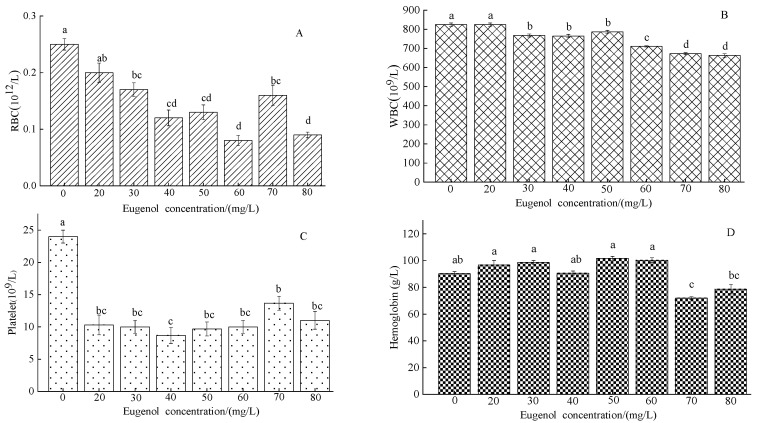
Changes in RBC, WBC, PLT, and HGB in crucian carp anesthetized with different concentrations of eugenol. The same lowercase letter in the graph indicates an insignificant difference (*p* > 0.05), whereas different lowercase letters indicate a significant difference (*p* < 0.05). Note: (**A**)—RBC, (**B**)—WBC, (**C**)—PLT, (**D**)—HGB.

**Figure 2 foods-12-02820-f002:**
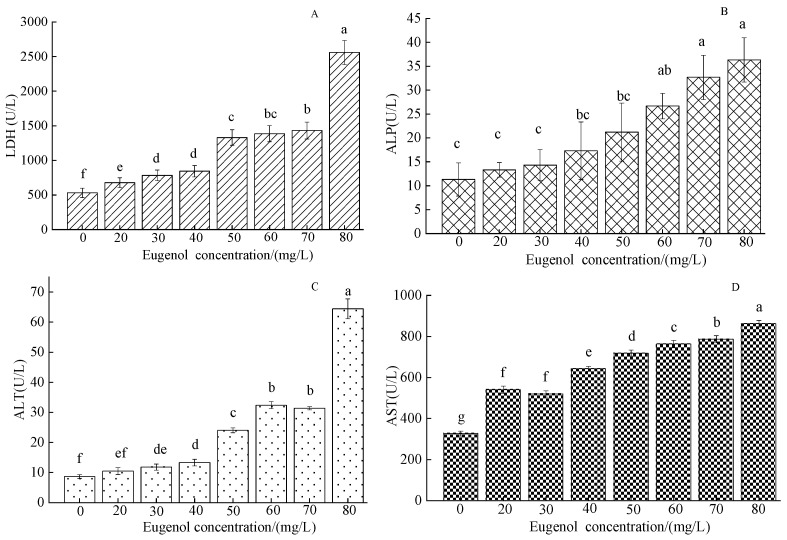
Changes in serum LDH, ALP, ALT, and AST content in crucian carp anesthetized with different concentrations of eugenol. Note: (**A**)—LDH, (**B**)—ALP, (**C**)—ALT, (**D**)—AST. The same lowercase letter in the graph indicates an insignificant difference (*p* > 0.05), whereas different lowercase letters indicate a significant difference (*p* < 0.05).

**Figure 3 foods-12-02820-f003:**
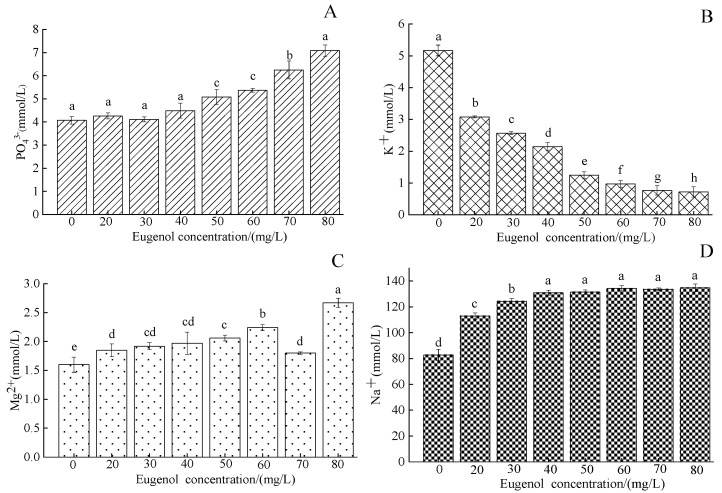
Changes in serum PO_4_^3−^, K^+^, Mg^2+^, and Na^+^ content in crucian carp after anesthesia with different concentrations of eugenol. Notes: (**A**)—PO_4_^3−^, (**B**)—K^+^, (**C**)—Mg^2+^, (**D**)—Na^+^. The same lowercase letter in the graph indicates an insignificant difference (*p* > 0.05), whereas different lowercase letters indicate a significant difference (*p* < 0.05).

**Figure 4 foods-12-02820-f004:**
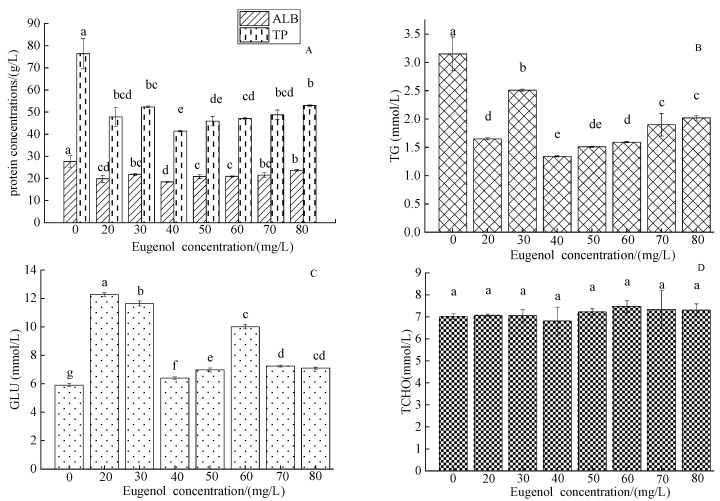
Changes in serum ALB, TP, TG, GLU, and TCHO in crucian carp after anesthesia with different concentrations of eugenol. Notes: (**A**)—ALB, TP, (**B**)—TG, (**C**)—GLU, (**D**)—TCHO. The same lowercase letter in the graph indicates an insignificant difference (*p* > 0.05), whereas different lowercase letters indicate a significant difference (*p* < 0.05).

**Figure 5 foods-12-02820-f005:**
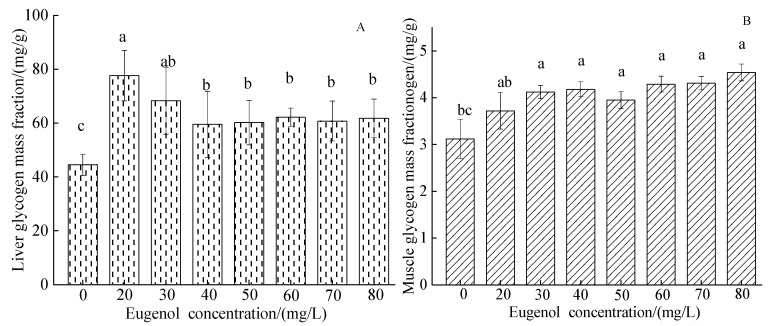
Changes in liver and muscle glycogen mass fraction content in crucian carp after anesthesia with different concentrations of eugenol. Notes: (**A**)—liver glycogen mass fraction, (**B**)—muscle glycogen mass fraction. The same lowercase letter in the graph indicates an insignificant difference (*p* > 0.05), whereas different lowercase letters indicate a significant difference (*p* < 0.05).

**Figure 6 foods-12-02820-f006:**
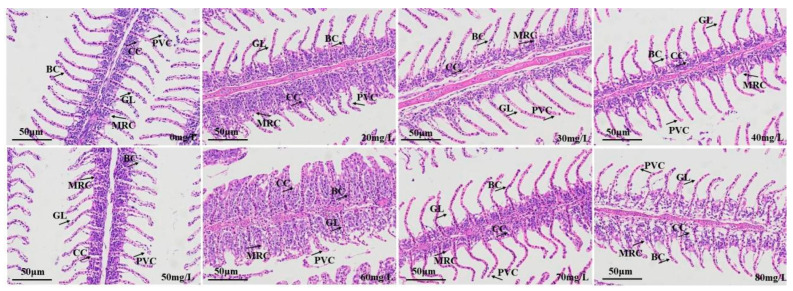
Effects of eugenol anesthesia on the histological structures in the liver in crucian carp. Note: BC: blood channel; GL: gill lamella; CC: chloride cell; PVC: pavement cells; MRC: mitochondrion-rich cell.

**Figure 7 foods-12-02820-f007:**
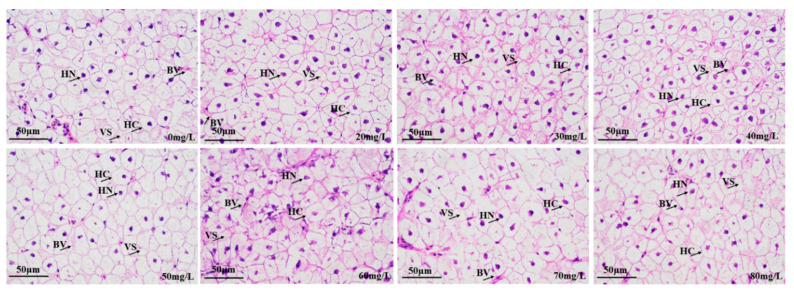
Effects of eugenol anesthesia on the histological structure of the liver in crucian carp. Note: VS: vacuoles; HC: hepatocytes; HN: hepatocyte nucleus; BV: blood vessels.

**Figure 8 foods-12-02820-f008:**
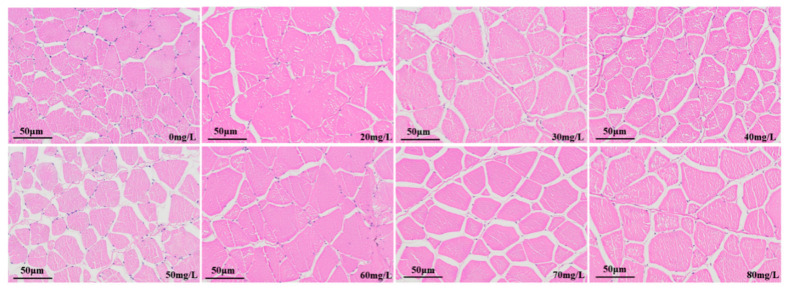
Effects of eugenol anesthesia on muscle structure in crucian carp.

**Figure 9 foods-12-02820-f009:**
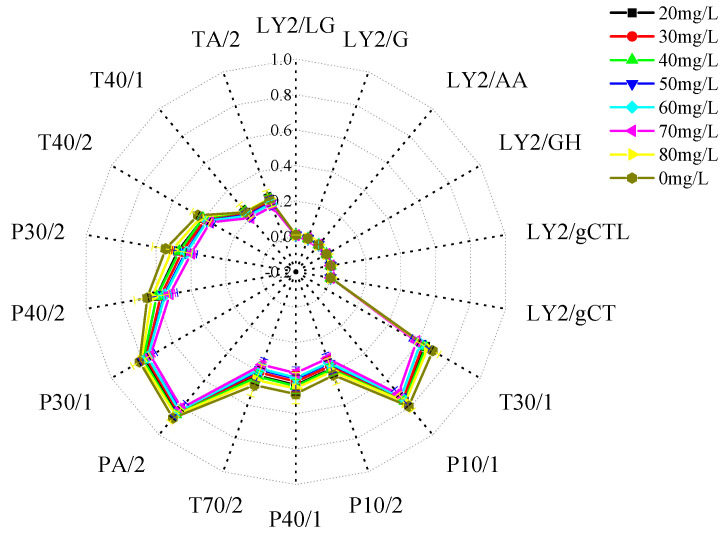
Radar fingerprinting of the effects of different concentrations of eugenol on muscle flavor in crucian carp.

**Figure 10 foods-12-02820-f010:**
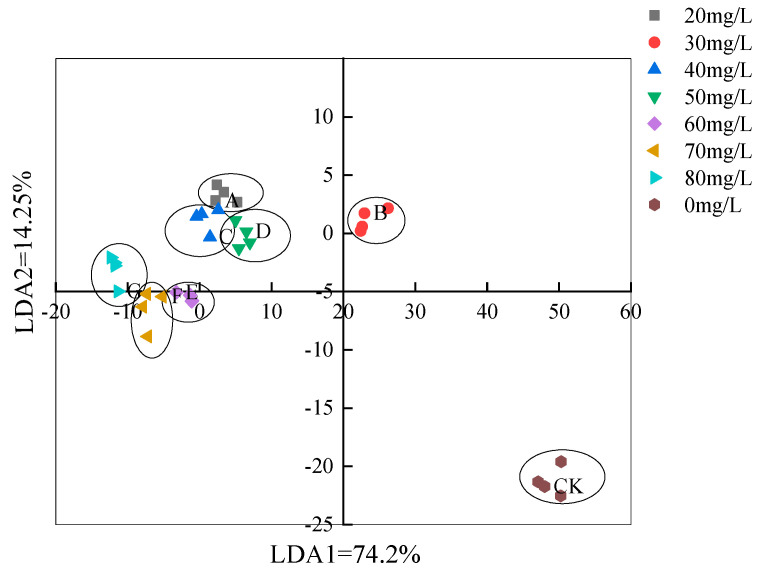
LDA analysis based on the electronic nose method. Note: CK: 0 mg/L, A: 20 mg/L, B: 30 mg/L, C: 40 mg/L, D: 50 mg/L, E: 60 mg/L, F: 70 mg/L, G: 80 mg/L.

**Table 1 foods-12-02820-t001:** Response characteristics of the sensors of the FOX4000 electronic nose.

Serial Number	Sensor Name	Sensor Response Characteristics
1	LY2/LG	Sensitive to gases with high oxidation capacity
2	LY2/G	Sensitive to toxic gases
3	LY2/AA	Sensitive to organic compounds
4	LY2/GH	Sensitive to toxic gases
5	LY2/gCTL	Sensitive to toxic gases
6	LY2/gCT	Sensitive to flammable gases
7	T30/1	Sensitive to organic compounds
8	P10/1	Sensitive to combustible gases
9	P10/2	Sensitive to flammable gases
10	P40/1	Sensitive to gases with high oxidation capacity
11	T70/2	Sensitive to aromatic compounds
12	PA/2	Sensitive to organic compounds, toxic gases
13	P30/1	Sensitive to combustible gases, organic compounds
14	P40/2	Sensitive to gases with high oxidation capacity
15	P30/2	Sensitive to organic compounds
16	T40/2	Sensitive to gases with high oxidation capacity
17	T40/1	Sensitive to gases with high oxidation capacity
18	TA/2	Sensitive to organic compounds

## Data Availability

The data used to support the findings of this study can be made available by the corresponding author upon request.

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
