# Peer review of "Comprehensive Analysis of Physiological, Biochemical and Flavor Characteristics Changes in Crucian Carp (Carassius auratus) under Different Concentrations of Eugenol"

_foods, 2023, doi:10.3390/foods12152820_

Round 1

Reviewer 1 Report

Dear Editor,

I studied the manuscript “Comprehensive analysis of Physiological, Biochemical and Flavor Characteristics changes in Crucian carp (Carassius auratus) under different concentrations of Eugenol”. This study aimed to investigate the effects of different concentrations of eugenol on blood physiological and biochemical indexes, muscle flavor, and histology in crucian carp. The study evaluated seven anaesthesia groups of 20, 30, 40, 50, 60, 70, and 80 mg/L and one control group (without eugenol) to determine the appropriate concentration of eugenol anesthetic for use in crucian carp transportation and production operations. The results showed that red blood cells and platelets of crucian carp decreased significantly with eugenol treatment, while white blood cells and hemoglobin did not change significantly. The study also found that eugenol treatment affected the flavor of the fish, but had little effect on the physiology and biochemistry of crucian carp in the concentration range of 20-80 mg/L. The comments to improve the scientific quality of the MS are given below:

-          Which is true? anesthesia or anaesthesia?

-          Trachinotus ovatus must be italic

-          Please spell DO

-          In the introduction, the importance of measuring physiological and biochemical indicators should be mentioned

-          Given that eugenol is dissolved in ethanol alcohol, was no treatment necessary as a positive control?

-          When exactly was the blood drawn? During 10 minutes? Or after 10 minutes?

-          How to kill fish as ethics should be mentioned.

-          Did the tanks containing 5 fish have repetitions? How many fish were used in total?

-          The separating the gills is not mentioned in the materials and methods.

-          Section 2.3 “In each experiment, when the fish into the complete anaesthesia period began timing, the complete anaesthesia period judgment standard for the experimental fish to the external stimulus loss of conditioned reflex ability, the bottom does not move, the fish was linear, and low frequency of gill movement.” is ambiguous and needs to be rewritten.

-          Five crucian carp were randomly placed in each tank, and no significant difference in fish size was observed among groups. Were the fish randomly placed in each tank or were the same size fish selected?

-          The titles of all subtitles are long and unconventional.

-          Shouldn't the acclimatization period be at least a week after transfer to the Research Center? While the experiment started only after 2 days.

-          The quality of photos and graphs is low

-          The first 9 lines of the discussion are appropriate for the Introduction

-          Did eugenol not affect chloride cells? And after that, it did not contribute to the absorption or excretion of sodium and potassium ions?

-          The following sentences are suitable for introduction:  “The gills are the main respiratory organ in fish, and are responsible for excreting metabolic wastes, such as ammonia nitrogen, and regulating osmotic pressure [55]. The fish gills are composed primarily of gill filaments and gill lamellae. The gill filaments extend vertically and have semicircular flat cystic gill lamellae, which are arranged in parallel. The main part of the gill filament contains gill cartilage, central venous sinus and gill filament epithelial cells (composed primarily of chloride cells, mucous cells, flat cells and undifferentiated cells). The gill lamellae are composed primarily of columnar cells, flat cells and blood vessels (with blood cells).”

-          The comparison of the effect of different concentrations of eugenol on different factors must be seen in the discussion, for example, the concentration of 20 and 80. Especially in cases where an abnormality has been seen in the amount of biochemical and physiological indicators.

-          Even though eugenol has caused a change in the flavor of the muscle, do you still recommend it in these concentrations? Why didn't you use concentrations lower than 20 for eugenol?

The quality of writing the English of the manuscript can be improved.

Author Response

Dear editor and Reviewers,

Thanks for your professional review work for foods-2492253. According to your suggestions, we have earnestly revised the manuscript. Here, the specific revisions are used in “Track Changes” in the manuscript and the answers are shown in detail as follows:

-         1、 Which is true? anesthesia or anaesthesia?

Response 1: Thank you for your good suggestion. I would prefer to use the British english "anaesthesia" instead of the American english "anesthesia". After all, "anaesthesia" is much closer to the greek word "αναισθησία" from which it derives.

-          2、Trachinotus ovatus must be italic

Response 2: Thank you for your good suggestion. Trachinotus ovatus has been changed to italic in the manuscript.

-        3、  Please spell DO

Response 3: Thank you for your good suggestion. The full name of DO is dissolved oxygen, which has been reflected in the manuscript.

-          In the introduction, the importance of measuring physiological and biochemical indicators should be mentioned

Response 4: Thank you for your good suggestion. The importance of measuring physiological and biochemical indexes has been introduced in discussion and conclusions of the manuscript. The importance of each physiological and biochemical indicators are introduced in detail in the discussion section. The introduction part aims to introduce crucian carp and eugenol.

-          Given that eugenol is dissolved in ethanol alcohol, was no treatment necessary as a positive control?

Response 5: Thank you for your good suggestion. In the research on eugenol as a fishery anaesthetic, ethanol was commonly used as a cosolvent to accelerate the solubility of eugenol in water, And there is no research data indicating that ethanol may affects eugenol anesthesia experiments.

-          When exactly was the blood drawn? During 10 minutes? Or after 10 minutes?

Response 6: Thank you for your good suggestion. The extraction of blood was completed immediately after the crucian carp entered the anaesthetic state for 10 minutes.

-          How to kill fish as ethics should be mentioned.

Response 7: Thank you for your good suggestion. All animal experiments were carried out in accordance with the U.K. Animals (Scientific Procedures) Act, 1986 and the National Institutes of Health Guide for the Care and Use of Laboratory Animals (NIH Publications No. 8023, revised 1978).

-          Did the tanks containing 5 fish have repetitions? How many fish were used in total?

Response 8: Thank you for your good suggestion. The tank with 5 fish is not repeated. Each fish is used only once. A total of 40 fish are used.

-          The separating the gills is not mentioned in the materials and methods.

Response 9: Thank you for your good suggestion. “Carefully open the operculum and cut the gills into 5-6 μm thick slices with scissors.”Added in the materials and methods.

-          Section 2.3 “In each experiment, when the fish into the complete anaesthesia period began timing, the complete anaesthesia period judgment standard for the experimental fish to the external stimulus loss of conditioned reflex ability, the bottom does not move, the fish was linear, and low frequency of gill movement.” is ambiguous and needs to be rewritten.

Response 10: Thank you for your good suggestion. The original statement is changed to “In each experiment, when the fish entered the complete anaesthesia period, the judgment standard of the complete anesthesia period was that the experimental fish lost the conditioned reflex ability to the external stimulation, the fish body tilted, did not struggle, the respiratory rate decreased but regular and the gill movement frequency was low.”

-          Five crucian carp were randomly placed in each tank, and no significant difference in fish size was observed among groups. Were the fish randomly placed in each tank or were the same size fish selected?

Response 11: Thank you for your good suggestion. The fish were randomly placed in each tank to select the fish of the same size.

-          The titles of all subtitles are long and unconventional.

Response 12: Thank you for your good suggestion. The titles of all subtitles have been revised in the manuscript.

-          Shouldn't the acclimatization period be at least a week after transfer to the Research Center? While the experiment started only after 2 days.

Response 13Thank you for your good suggestion. The meaning in the manuscript is not clearly expressed, and it is now expressed as“they were transported to the Research Center for temporary breeding for 2 weeks, continuously inflated, and fed with commercial compound feed once in the morning and once in the evening every day, and treated with sewage and water exchange. During this period, fish that died or survived in poor condition were removed. Feeding was prohibited 2 days before the start of the experiment.”

-          The quality of photos and graphs is low

Response 14Thank you for your suggestion, but the picture of the experiment is the software output high-definition picture.

-          The first 9 lines of the discussion are appropriate for the Introduction

Response 15Thank you for your good suggestion. The first 9 lines of the discussion is to briefly introduce the effect of various indicators, so as to lead to the introduction of follow-up test results.

-          Did eugenol not affect chloride cells? And after that, it did not contribute to the absorption or excretion of sodium and potassium ions?

Response 16Thank you for your good suggestion. The results showed that eugenol had no significant effect on chloride cells and sodium ions in gill tissue of crucian carp, but had a significant effect on potassium ion content.

-          The following sentences are suitable for introduction: “The gills are the main respiratory organ in fish, and are responsible for excreting metabolic wastes, such as ammonia nitrogen, and regulating osmotic pressure [55]. The fish gills are composed primarily of gill filaments and gill lamellae. The gill filaments extend vertically and have semicircular flat cystic gill lamellae, which are arranged in parallel. The main part of the gill filament contains gill cartilage, central venous sinus and gill filament epithelial cells (composed primarily of chloride cells, mucous cells, flat cells and undifferentiated cells). The gill lamellae are composed primarily of columnar cells, flat cells and blood vessels (with blood cells).”

 Response 17Thank you for your good suggestion. The following sentences “the gills are the main respiratory organ in fish, and are responsible for excreting metabolic wastes, such as ammonia nitrogen, and regulating osmotic pressure [55]. The fish gills are composed primarily of gill filaments and gill lamellae. The gill filaments extend vertically and have semicircular flat cystic gill lamellae, which are arranged in parallel. The main part of the gill filament contains gill cartilage, central venous sinus and gill filament epithelial cells (composed primarily of chloride cells, mucous cells, flat cells and undifferentiated cells). The gill lamellae are composed primarily of columnar cells, flat cells and blood vessels (with blood cells).”In the discussion and summary section, we want to introduce the importance of each index first, and then discuss and analyze the results of different indexes combined with the test results.

-          The comparison of the effect of different concentrations of eugenol on different factors must be seen in the discussion, for example, the concentration of 20 and 80. Especially in cases where an abnormality has been seen in the amount of biochemical and physiological indicators.

Response 18Thank you for your good suggestion. The changes of physiological and biochemical indexes of different concentrations of eugenol on the anaesthetic effect of crucian carp have been analyzed in the results. 

-    Even though eugenol has caused a change in the flavor of the muscle, do you still recommend it in these concentrations? Why didn't you use concentrations lower than 20 for eugenol?

 Response 19Thank you for your good suggestion. Although eugenol caused changes in muscle flavor, considering the long-term development of fishery anesthetics, it is possible to study fishery anesthetics that have no effect on muscle. The selection criteria of the anesthetic concentration of eugenol in this experiment is based on the results of some previous studies. In addition, in the previous experiment, the anaesthetic effect of eugenol concentration below 20mg/L on crucian carp is not obvious.

Finally, we would like to thank you again and we look forward to receiving your suggestions.

Best regards,

First author: Lexia Jiang

E-Mail: 13977695205@163.com

Corresponding author: Changfeng Zhang

E-Mail: zcf202@163.com

Reviewer 2 Report

Overall

-          Please use the template with line number.

-          I think the authors were studying the acute effects of eugenol on Crucian carp, as the samples were collected almost immediately after eugenol exposure. That should be reflected on the title and the content of the manuscript.

Abstract

1.       “triglyceride first increased and subsequently de-creased (P<0.05); blood glucose content first increased and then decreased (P<0.05)”It is rather confusing. Did the authors conduct repeated sampling on the fish?

2.       “The anaesthesia concentration of 20–80 mg/L had little effect on the physiology and biochemistry of crucian carp...” With the results provided by the authors, I do not agree with that. Even with single exposure, the authors showed results that the eugenol led to gill and kidney damages (possible liver damage as well), decreased hematological indices etc. Modification will be needed.

Introduction

3.       “As a food additive, eugenol is usually used as a dental analgesic in China.” Eugenol do used as a food additive for farmed animal, but not a food additive for human. Would the authors clarify the meaning?  

4.       The author should provide details on the following issues, to help the future reader to understand the situation:

Is it a common practice to use eugenol/clove oil throughout the supply chain?

Is it legal to use eugenol in food fish in China? I think the authors should indicate that to

What is the common practice of using eugenol?

Materials and methods

5.       Even though a reference is cited, the blood sampling process is not clearly described. How exactly the blood samples were collected? What needle was used? How many (volume) samples were collected? Was there any pooling? What anticoagulant was used?

6.       Did the blood samples, liver samples and gill samples from the same fish?

Discussion

7.       “In general, the permeability of fish gill epithelium was greatest to hydrogen...”. Do the author mean hydrogen ions?

8.       “Total cholesterol is not only an important component of biofilm but also a prerequi-site for the synthesis of bile acids...”. This sentence is odd. Biofilm is the colonization of microorganisms (e.g. bacteria) on the solid surface. What is the relationship of cholesterol and biofilm?

9.       For your histology part, how many fields under microscope did the author observed to draw the conclusion?

10.    “When the content of ALT and AST in the serum increases, hepatocytes are considered to be damaged, and ALT is an important indicator of liver damage.” The discussion on liver is contradictory. On one hand, the authors mentioned that the increase in ALT and AST is an indicator of liver damages, but then try to provide a reference to justify it was normal when exposed to eugenol. However, in the reference the author cited [38], the changes in ALT and AST appeared in the recovery stage, not during the exposure. Therefore, the conclusion the author made here might not be that accurate.

11.   The authors only discussed the changes of gill after exposing to 60 mg/L eugenol. However, with the figures provided in the manuscript, fish exposed to other dosage of eugenol also showed obvious changes. More discussion should be provided. Also, in the reference 38, those authors tried to categorize the damage of organs after exposure to different chemicals. Would the authors attempt that as well?

12.   In p.12, the authors suggested “We speculated that with increasing eugenol concentration, the damage of anaesthetic to the fish kidneys caused by the anesthetic was aggravated, thus resulting in renal dysfunction, abnormal magnesium metabolism, serum magnesium retention and increased level.” And then, in p. 13, the authors mentioned “In this study, the different eugenol anaesthesia concentrations in crucian carp did not cause liver tissue damage,” which are also contradictory. Please write according to your observation.

13.   “The gills are involved in respiration, acid-base balance, ion balance, nitrogenous waste excretion, osmotic adjustment and other multi-functional tissues...” I think “gaseous exchange” is more accurate to describe the function of fish gill

Conclusion

Please update according to the changes you have made in the main text.

Except for some vocabulary, the overall writing is acceptable. 

Author Response

Dear editor and Reviewers,

Thanks for your professional review work for foods-2492253. According to your suggestions, we have earnestly revised the manuscript. Here, the specific revisions are used in “Track Changes” in the manuscript and the answers are shown in detail as follows:

Abstract

  1. “triglyceride first increased and subsequently de-creased (P<0.05); blood glucose content first increased and then decreased (P<0.05)”It is rather confusing. Did the authors conduct repeated sampling on the fish?

Response 2: Thank you for your suggestion. In this experiment, each index was tested repeatedly, and each fish was tested only once.

  1. “The anaesthesia concentration of 20–80 mg/L had little effect on the physiology and biochemistry of crucian carp...” With the results provided by the authors, I do not agree with that. Even with single exposure, the authors showed results that the eugenol led to gill and kidney damages (possible liver damage as well), decreased hematological indices etc. Modification will be needed.

Response 3: Thank you for your suggestion. In this experiment, the experimental results showed that the anaesthetic concentration of 20 ~ 80 mg/L had a great influence on PO43-, Mg2+ and K+ of crucian carp, but had little effect on the Na+ content and liver tissue morphology, and caused slight swelling of the gill lamellae on the gill tissue. The original sentence “The anaesthesia concentration of 20-80 mg/L had little effect on the physiology and biochemistry of crucian carp...”has been changed to “The anaesthesia concentration of 20-80 mg/L had some effect on the physiology and biochemistry of crucian carp... ” and has been reflected in the manuscript.

Introduction

  1. “As a food additive, eugenol is usually used as a dental analgesic in China.” Eugenol do used as a food additive for farmed animal, but not a food additive for human. Would the authors clarify the meaning?

Response 4: Thank you for your suggestion. Whether eugenol is used as an additive in animal or human food is not the focus of this manuscript. In order to avoid misunderstanding, we delete the sentence “as a food additive”.

  1. The author should provide details on the following issues, to help the future reader to understand the situation:

Is it a common practice to use eugenol/clove oil throughout the supply chain?

Response 5: Thank you for your suggestion. This manuscript mainly studies the effects of eugenol at the concentration of 20 ~ 80 mg/L on the blood physiology and biochemistry and muscle flavor of crucian carp. This manuscript does not involve how eugenol is used during transportation.

Is it legal to use eugenol in food fish in China? I think the authors should indicate that to.

Response 6: Thank you for your suggestion. The main research for eugenol is the anaesthetic effect and safety. The purpose is to evaluate whether eugenol could be used as a fishery anesthetic in the future.

What is the common practice of using eugenol?

Response 7: Thank you for your suggestion. According to the current research shows that more is in the study of the function of eugenol, less in the practical application and introduction. .

Materials and methods

  1. Even though a reference is cited, the blood sampling process is not clearly described. How exactly the blood samples were collected? What needle was used? How many (volume) samples were collected? Was there any pooling? What anticoagulant was used?

Response 8: Thank you for your suggestion. Five test fish were selected at each blood sampling and quickly fished out of the tank. Using a 5 mL disposable sterile syringe to draw blood from the tail vein of crucian carp, the blood volume is more than 5 mL, and each fish only draws blood once. After the blood samples were placed in a refrigerator at 4 °C for 10 min, some of them were added with EDTA anticoagulant to prevent coagulation, which was used for the determination of blood physiological indexes. The other part was centrifuged at 4 °C, 4 000 r / min for 20 min, and the supernatant was stored in the refrigerator at -80 °C for the determination of serum biochemical indexes. It has been revised in the manuscript.

  1. Did the blood samples, liver samples and gill samples from the same fish?

Response 9: Thank you for your suggestion. Anaesthesia with different concentrations of eugenol blood samples, liver samples and gill samples of different fish were taken, and repeated tests were performed at each concentration.

Discussion

  1. “In general, the permeability of fish gill epithelium was greatest to hydrogen...”. Do the author mean hydrogen ions?

Response 10: Thank you for your suggestion. “In general, the permeability of fish gill epithelium was greatest to hydrogen ions.” It has been revised in the manuscript.

  1. “Total cholesterol is not only an important component of biofilm but also a prerequi-site for the synthesis of bile acids...”. This sentence is odd. Biofilm is the colonization of microorganisms (e.g. bacteria) on the solid surface. What is the relationship of cholesterol and biofilm?

Response 11: Thank you for your suggestion. In addition to controlling the fluidity of the membrane, cholesterol is also a precursor of many important active substances in the body. Sex alcohol hormones, vitamin D, and bile acids are derivatives of cholesterol. Cholesterol plays an important role in regulating membrane fluidity, increasing membrane stability and reducing the permeability of water-soluble substances.

  1. For your histology part, how many fields under microscope did the author observed to draw the conclusion?

Response 12: Thank you for your suggestion. The conclusions drawn from 6 different fields of view were observed under the microscope, and an optimal field of view combination was selected from each concentration to draw conclusions.

  1. “When the content of ALT and AST in the serum increases, hepatocytes are considered to be damaged, and ALT is an important indicator of liver damage.” The discussion on liver is contradictory. On one hand, the authors mentioned that the increase in ALT and AST is an indicator of liver damages, but then try to provide a reference to justify it was normal when exposed to eugenol. However, in the reference the author cited [38], the changes in ALT and AST appeared in the recovery stage, not during the exposure. Therefore, the conclusion the author made here might not be that accurate.

Response 13: Thank you for your suggestion. Originally cited [38], the changes in ALT and AST appeared in the recovery stage, not during the exposure, but in this manuscript, the changes in ALT and AST content after exposure to eugenol were studied, so other references were selected.

  1. The authors only discussed the changes of gill after exposing to 60 mg/L eugenol. However, with the figures provided in the manuscript, fish exposed to other dosage of eugenol also showed obvious changes. More discussion should be provided. Also, in the reference 38, those authors tried to categorize the damage of organs after exposure to different chemicals. Would the authors attempt that as well?

Response 14: Thank you for your suggestion. In the newly cited [38], the research ideas and methods are similar to this manuscript.

  1. In p.12, the authors suggested “We speculated that with increasing eugenol concentration, the damage of anaesthetic to the fish kidneys caused by the anesthetic was aggravated, thus resulting in renal dysfunction, abnormal magnesium metabolism, serum magnesium retention and increased level.” And then, in p. 13, the authors mentioned “In this study, the different eugenol anaesthesia concentrations in crucian carp did not cause liver tissue damage,” which are also contradictory. Please write according to your observation.

Response 15: Thank you for your suggestion. The experimental results showed that the levels of PO43- and Mg2+ after eugenol anaesthesia were higher than those in the control group. Studies have shown that the kidney is related to the metabolism of magnesium and phosphorus. When the kidney is damaged, the level of magnesium and phosphorus metabolism in the fish will be abnormal. It is speculated that the high concentration of eugenol causes kidney damage; however, from the observation of HE stained liver tissue, different concentrations of eugenol anaesthesia did not cause liver tissue damage.

  1. “The gills are involved in respiration, acid-base balance, ion balance, nitrogenous waste excretion, osmotic adjustment and other multi-functional tissues...” I think “gaseous exchange” is more accurate to describe the function of fish gill

Response 16: Thank you for your suggestion. The respiration has been replaced by gaseous exchange in the manuscript. 

Finally, we would like to thank you again and we look forward to receiving your suggestions.

Best regards,

First author: Lexia Jiang

E-Mail: 13977695205@163.com

Corresponding author: Changfeng Zhang

E-Mail: zcf202@163.com
